# Interaction between Cardiac Functional Indices during Incremental Exercise Test Reveals the Peculiarities of Adaptation to Exercising

**DOI:** 10.3390/medicina55070314

**Published:** 2019-06-26

**Authors:** Deivydas Velicka, Zivile Kairiukstiene, Kristina Poderiene, Alfonsas Vainoras, Jonas Poderys

**Affiliations:** 1Department of Sport Medicine, Lithuanian University of Health Sciences, 47181 Kaunas, Lithuania; 2Institute of Sport Science and Innovations, Lithuanian Sports University, 44221 Kaunas, Lithuania

**Keywords:** exercising, stroke volume, cardiac output, interaction, adaptation

## Abstract

*Background and objectives*: Physical load causes structural changes in the heart that vary depending on the type of training and may affect the function of the heart. Aim of the study: To determine, using the applied co-integration method on algebraic data, the impact of sprinting and of endurance adaptation on the dynamic interactions of cardiovascular functional indices while participants were performing under an increasing workload, up to their inability to continue. *Materials and Methods:* Healthy athletes were chosen to take part in this study and were separated into two groups: Sprinters (*n* = 11) and endurance athletes (*n* = 13). The bicycle ergometric method of incremental increase in a provocative workload (graded stress) was used. The heart rate, stroke volume, and cardiac output were determined using the tetrapolar rheography method. *Results:* Individuals who are adapted to endurance while carrying physical loads, in contrast to well adapted sprinters, are characterized with a lower rate of changing the pace of interactions between stroke volume and cardiac output while performing at an increasing workload up to their inability to continue. Also, endurance athletes displayed a long and relatively stable phase as well as a greater decrease of interaction between indices at the end of the workload. At the beginning of the exercise, the interaction between the stroke volume and the cardiac output was reduced. However, as the physical load continued, this interaction became significantly stronger. The comparison of the stroke volume and the cardiac output’s dynamic interaction revealed that the endurance group had a greater working capacity. *Conclusions:* Typical dynamics of interactions during the testing with an increasing physical load can be differentiated into separate phases: The decrease of interaction at the onset of the load, the increase during the continuation of an increased workload, and the decrease at the last stages of the load.

## 1. Introduction

Functional alterations in the human body during exercise are conditioned by a sequence of complex interrelated processes. These processes also cause a physiological adaptation to physical loads, which is generally defined as a development of such functionality, and which would manifest itself in the individual’s greater tolerance to physical workloads, an increased amount of energy resources, and an optimization of regulatory mechanisms [1]. Human myocardium’s structural and functional changes induced by physical loads are also evidence of a long-term adaptation. Myocardium, unlike skeletal muscle, is fatigue resistant during long-term exercise. However strenuous exercises cause heart fatigue [2] and potentially can induce heart-tissue damage [3]. Therefore, distinguishing exercise-induced adaptation peculiarities requires the consideration of intrinsic cardiac function and its structure.

Factors such as duration or intensity of exercise [4], training status [5], gender [6], and environment [7] are indistinguishable components that determine the extent of post-exercise dysfunctions [8]. Furthermore, physical exertion that leads to structural cardiac changes vary depending on the type of training and may affect the function of the heart, i.e., long-term physical activity may lead to an athlete developing “athlete’s heart” [9]. Athletes’ hearts adapt to the increasing physical load by increasing stroke volume. Since stroke volume is an essential structural component of cardiac output, an exercise-induced increase of cardiac output is caused by an increase of the heart rate and of the stroke volume. Lee and Oh [10] confirmed, in their study involving 22 subjects with at least five or more years of permanent training (swimming), that physical exercise has a positive effect on the heart’s structural changes, but there is no significant impact on its function.

Such conflicting views encouraged us to investigate to see if relevant information about physiological processes, in terms of long-term adaptation, could be obtained by estimating and analyzing the most relevant indicators and their dynamic interrelations when one of the indicators is another’s structural component. For this reason, we hypothesized that a stroke-volume decrease at the onset of a physical load would be minor and that an increase of stroke volume would emerge promptly. This phenomenon should be a reasonably accurate indication of a person’s better physical condition when carrying out a long physical workout. In order to confirm or reject our hypothesis, we used the model of the human body that concerns the synergetic aspects between the systems’ components [1]. This model assisted us in assessing the indicators of cardiovascular system and to reveal their hemodynamic characteristics when evaluating their interactions to see which is a great contribution to improving the evaluation precision of the cardiovascular system functions. In this study, our main purpose was to determine the impact of sprinting and endurance adaptation on the dynamic interactions of cardiovascular functional indices while a person was performing an increasing workload, up to their inability to continue.

## 2. Materials and Methods

### 2.1. Participants

The contingent of subjects consisted of 24 healthy athletes with at least six years of permanent training experience. Subjects were assigned to one of two groups: Endurance (n = 13, age 26.1 ± 1.7 years, height 178 ± 3.7 cm, body mass 74.8 ± 4.1 kg) and sprinters (n = 11, age 27.3 ± 3.2 years, height 177.6 ± 3.8 cm, body mass 77.8 ± 5.2 kg). There was no significant difference between the cohorts.

All subjects gave their informed consent for inclusion before they participated in the study. The study was conducted in accordance with the Declaration of Helsinki, and the protocol was approved by the Regional Biomedical Research Ethics Committee (Lithuanian University of Health Sciences, Kaunas, Lithuania) (No: BE-2–38, 07–09-2016).

### 2.2. Organization of the Study

The bicycle ergometric method of incremental increase in a provocative workload (graded stress) was used. The subjects underwent a 50 W increase in workload every 6 min (60 revolutions/min), and they exercised up to their inability to continue the task or to a predetermined goal (submaximal heart rate) unless distressing cardiovascular symptoms supervened. After the physical load the subjects rested and the cardiac parameter registered 5 min. Participants during all testing procedures were seated on a bicycle.

### 2.3. Measurements

At the end of the 2nd; 4th and 6th minutes of each step of the workload, the subjects were asked to stop exercising for five to six seconds in order for us to measure the stroke volume and cardiac output with tetra polar chest rheography (RPG2-02, Medtekhnika, Moscow, Russia), a convenient and non-invasive method based on the Kubicek procedure [11]. To achieve this, we used a four-electrode impedance plethysmographic system. Two-band electrodes were placed around the base of each subject’s neck; the third band electrode was placed around his thorax, 2 cm below the xiphisternal joint; and the fourth band electrode was placed around the abdomen, like a belt. The two outer electrodes were placed about 2 cm away from the inner electrodes. The upper neck and abdomen electrodes were excited with a constant 100 kHz sinusoidal current and the resultant voltage (impedance) changes that occurred in the cardiac cycles were monitored from the two inner electrodes. The stroke volume was calculated from the impedance change information using a formula that relates impedance changes to volume changes in a conducting solid [12].

### 2.4. Mathematical Methods

Statistical approach based on averaged data analysis cannot describe the synergetic interactions between the physiological system and mechanisms while exercising [13,14]. A recently developed algebraic co-integration method for the measurement of interactions between two physiological parameters was used [15]. According to this approach, two synchronous signals are taken at discrete time intervals and after normalization. Normalization (to interval [0; 1]) is required for the correct interrelations analysis and to satisfy the equation where the normalized value is equal to the difference between the original value and the minimal physiological value, divided by the difference between the maximal and minimal physiological values. These signals are co-integrated into the second order matrix. From the initial parameters of the matrix (difference dfr A_n_ = x_n_ − y_n_ and co-diagonal product cdp A_n_ = ab(x_n − 1_ − y_n − 1_)(x_n + 1_ − y_n + 1_), in both cases x_n_ and y_n_ are real numbers, and they represent the recorded parameters following characteristics that have a more comprehensive sense, i.e., discriminant (Dsk): Dsk A_n_ = (dfr A_n_)^2^ + 4 cdp A_n_. Large Dsk values indicate low inter-parametric concatenation, whereas small Dsk values (close to zero) correspond to a high interaction between the analyzed parameters.

### 2.5. Statistical Analysis of the Data

The Kolmogorov–Smirnov test was used to determine whether the data had an underlying normal distribution, and the required homogeneity variance of compared samples was verified with Levene’s test. ANOVA followed by Tukey post-hoc analyses were used to compare the statistical significance of the differences between the groups and for multiple comparisons within each group. The calculation for the observed power was performed, and the partial eta squared (ŋ^2^) was estimated as a measure of the test-retest effect size. The difference was regarded as statistically significant if *p* < 0.05. The values are reported as arithmetical mean ± standard error of the mean (SEM). All data analyses were performed using SPSS version 21.0 (IBM Corporation, Armonk, NY).

## 3. Results

Cardiovascular parameters and their dynamic interactions comparing the sprinters and the endurance groups are presented in Figure 1, Figure 2 and Figure 3. Alterations in heart rate during the applied testing illustrate typical behavior of self-organizing systems, depending on the type of adaptation, i.e., sprinting or endurance. The heart rate changes of both groups while performing stepwise increasing workloads up to inability are presented in Figure 1.

When we compared recorded initial heart rate values that were in a relative state of rest, i.e., we found no statistically significant difference between the groups during exercise. Workload-induced heart-rate increase was more acute in the sprint group at each stage of the incremental load, and during the 150 W and 200 W workload, the difference between the groups reached a statistically significant level (*p* < 0.05). However, comparing the registered subjects’ maximum heart rate values during the last stage of workload, we did not detect a statistically significant difference between the groups.

Figure 2 shows the stroke volume and cardiac output variations in the two groups while the subjects performed under increasingly more intense physical loads.

The endurance group was characterized by higher stroke volume values before testing (endurance group – 89.1 ± 5.1 mL, sprinters’ group—74.7 ± 6.7 mL); however, this was not a statistically significant difference. The stroke volume increased in parallel with the intensity of the workload, but significantly higher stroke-volume values were observed in the endurance subjects at the level of 150 W (endurance group—130.8 ± 7.8 mL, sprint group—105.1 ± 7.1 mL; *p* < 0.05) and also at the level of 200 W or while comparing maximum stroke-volume values (*p* < 0.05). Figure 2B shows that the cardiac output increased stepwise, i.e., the parameter increased at the onset of the first minute of a new stage of the physical load and during the five minutes that remained at the same level. Such a gradual increase lasted until the load reached 250 W. A more pronounced increase of cardiac output was observed in the endurance group at the stages of 200–250 W, and this group was characterized by significantly higher cardiac output values, as recorded in the last stages of the workload (*p* < 0.05).

Figure 3 displays the dynamics of the interaction between stroke volume and cardiac output while the subjects performed stepwise increasing workloads up to their inability to continue in both groups.

Various peculiarities of Dsk changes were determined by comparing the initial Dsk (stroke volume, cardiac output), and the degree and the duration of this increase between the groups (Figure 3A). Before the testing of the state of relative rest, the calculated Dsk values were lower in the sprinters’ group (0.35 ± 0.06) than in the endurance group (0.37 ± 0.04), but the difference between groups was not statistically significant.

The difference between groups was determined by analyzing the initial Dsk, its growth rate and the duration of the growth (Figure 3B). The Dsk of the endurance group increased within the first five minutes of the physical load (0.46 ± 0.06), and, compared with the initial Dsk values registered before the load, a statistically significant difference was obtained (*p* < 0.05). Afterwards it began to decline, i.e., concatenation between the stroke volume and cardiac output began to increase. The sprinters’ group’s increase in Dsk values for the same concatenation lasted longer (until the 7th min of the load) and increased up to 0.61 ± 0.07, which was significantly higher (*p* < 0.05) than in the endurance group.

The subjects’ Dsk values, for both groups, while they were performing at the increasing load in the later stages of the bicycle ergometer testing, were constantly declining. The sprinters’ group’s Dsk values decreased up to approximately the 25th minute of testing, reaching 0.13 ± 0.04, and afterwards the values began to increase. The endurance group was characterized by exactly the same Dsk change trend as the sprinters’ group; however, the decline of Dsk lasted significantly longer (*p* < 0.05). While they were overcoming the last subjectively difficult stages of physical load, their Dsk levels increased less.

During the recovery after the workload on the bicycle ergometer, the Dsk values (Figure 3A) also increased; however, within four minutes of recovery, the values did not reach the baseline (0.20 ± 0.04). Nevertheless, the recovery process for the endurance group was faster than for the sprinters’ group. Unlike the sprinters’ group, the endurance group’s results indicate that a statistically significant difference (*p* < 0.05) was revealed in the Dsk values between the last load stage and the recovery period (0.13 ± 0.01 and 0.29 ± 0.04, respectively).

## 4. Discussion

Dynamic systems theory applied in human physiology and sports medicine provides the ability to explore the functional system’s behavior and to provide information about the system’s organization [16,17]. This research work was focused on the adaptation impact on the cardiovascular indices and their interrelationships that are responsible at the highest level for the homeostasis of the human body during intensive physical loads. Attention was also paid to the body’s self-organization during an incremental workload.

A cardiac-output increase induced by a physical load is conditioned by an increase in the heart rate and in the stroke volume [18]. Therefore, stroke volume is an essential structural component of cardiac output. A decrease of the stroke volume and of the cardiac output concatenation manifested at the onset of a physical load, i.e., Dsk values increased in both groups of subjects. At the onset of the physical load, the interaction between stroke volume and cardiac output became weaker, i.e., Dsk values increased, and this was the case for both groups of subjects as well as for each subject in general.

During the bicycle ergometer test, an exercise-induced Dsk increase was observed; however, this change was not fast. The estimated Dsk between stroke volume and cardiac output and its changes show that both at rest and during the physical load, the two parameters are strongly interrelated. Only at the stage of the subjectively intensive workload did the Dsk values begin to increase.

Interestingly, the Dsk values decreased at a slower pace for the endurance athletes than for the sprinters; however, during the recovery phase, the endurance group exhibited a significantly faster increase in Dsk than the sprinter athletes. We can possibly relate this to the idea of a muscle being a defined chaos-system function that attempts to recover as fast as possible even within small ranges of power reduction. This allows us to arrive at a conclusion that an interruption to recovery processes in endurance sports players for any reason (such as adaptive processes becoming disjointed) can severely deteriorate athletes’ results.

According to the results of Zumbakyte-Sermuksniene et al. [19], where the effect of basketball players’ anthropometric data of their cardiovascular parameter alterations during a physical load were analyzed, the regulatory systems of recruited taller and heavier male basketball players tended to load more slowly and more economically than did those of their shorter and lighter counterparts. The interrelation of the heart rate in association with other cardiovascular parameters represents the processes of the regulatory system and also has a direct impact on cardiac-output dynamics.

Obscured Dsk values had an upward trend in both groups up to the 25th minute of physical load. Then the Dsk-value increases accelerated, and it took much longer for the endurance than for the sprinters’ group, i.e., with increasing Dsk values, individuals who adjusted to endurance workloads were able to continue to exercise longer. The greatest working capacity was shown by the endurance group, who continued to work for 39 min, in contrast to the sprint group who had a 27 min average. In the run-up to the physiological limit, greater “autonomy” of the heart’s regulative systems is allowed in order to optimize the potential solutions in a critical situation, i.e., to find the best approach for the recruitment of all compensating mechanisms to meet the body’s need to maintain an adequate hemodynamic.

The long-lasting changes of concatenation among registered indices commenced at the onset of exercising, and this type of change was observed during workloads of increased intensity until the subjects reached a specific functional state (fatigue) when the increase of concatenation had changed to the contrary. This finding suggests that changes of interactions between the indices, which represent cardiac performance, appear at the onset of exercising and that the repeatedly emerged decrease of interaction in the period of maximal physical load leads to the inability to continue exercising.

Therefore, the algebraic data co-integration method for the evaluation of biological process dynamics provides access to qualitatively new information about the body’s functional indicators’ change in terms of their diversity and generality at the same time. At the onset of physical load, the stroke volume and cardiac output dynamic interaction is weaker, and, if physical activity is continued, this concatenation gets increasingly stronger. Such changes of stroke volume and cardiac output concatenation to a lesser degree were more common to the endurance cohort. Athletes with different adaptations to load (endurance or sprinters) have differing rates of activating their cardiac metabolism. Rowe, Safdar, and Arany [20] have shown that long-term endurance training leads to a greater working economy: When a subject is at the same absolute running speed and intensity, less oxygen is consumed. Therefore, individuals who are adapted to endurance workloads might be distinguished for their lower metabolic rates and for their enhanced myocardial economy compared to their sprinting peers. Reger et al. [8] have also confirmed that there is a temporal component to the physiological stress of exercise, which also contains an increased susceptibility to myocardial ischemic injury and might be metabolic in nature.

One of the main limitations of this investigation is the fact that it is difficult to interpret this kind of data due to the lack of studies of this type. Although some studies with the methodology we used have been published [1,15], there is still a lack of determined standards that could be used during this kind of data analysis. Also, in such data analyses, the initial data is susceptible to discrimination; thus, the change trends and the moments at which the contrary shifts emerge should be emphasized. Nevertheless, the results we obtained during this study in which we applied recently developed methodology allowed us to differentiate some specific features of the dynamics of cardiac performance that were affected by increasing fatigue based on the type of adaptation.

## 5. Conclusions

Typical dynamics of interactions during testing with an increasing physical load can be differentiated into separate phases: The decrease of interaction at the onset of a load; the increase during the continuation of an increased workload, and the decrease at the last stages of the load. The decrease of interactions is more exposed among the endurance cohorts. During exercise, interactions between stroke volume and cardiac output become significantly stronger; however, development of fatigue is followed by a decrease in this interaction (an increase of Dsk). Different types of sports players (endurance athletes or sprinters) have different interconnection behaviors in their bodies. The changes in the typical dynamic of interconnections as well as in the adaptability to load and in recovery processes can reveal possible problems in adaptability to load.

## Figures and Tables

**Figure 1 medicina-55-00314-f001:**
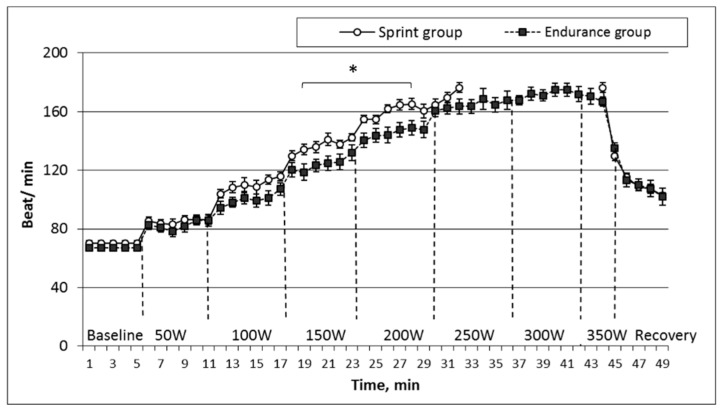
Changes of heart rate in sprint and endurance groups while performing incremental increase in workload every 6 min.

**Figure 2 medicina-55-00314-f002:**
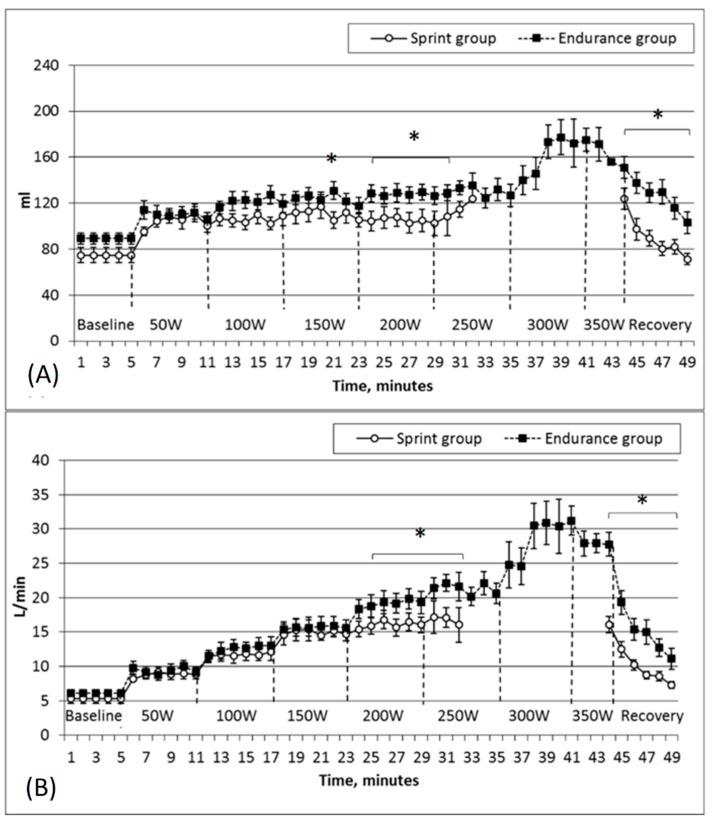
Changes of stroke volume (**A**) and cardiac output (**B**) in sprint and endurance groups while performing under an incremental increase in workload every 6 min. Note: Axis x show the study protocol, i.e., the increase of workload in watts and at 44–49 min—the change of parameter during the recovery. * Statistically significant difference (*p* < 0.05).

**Figure 3 medicina-55-00314-f003:**
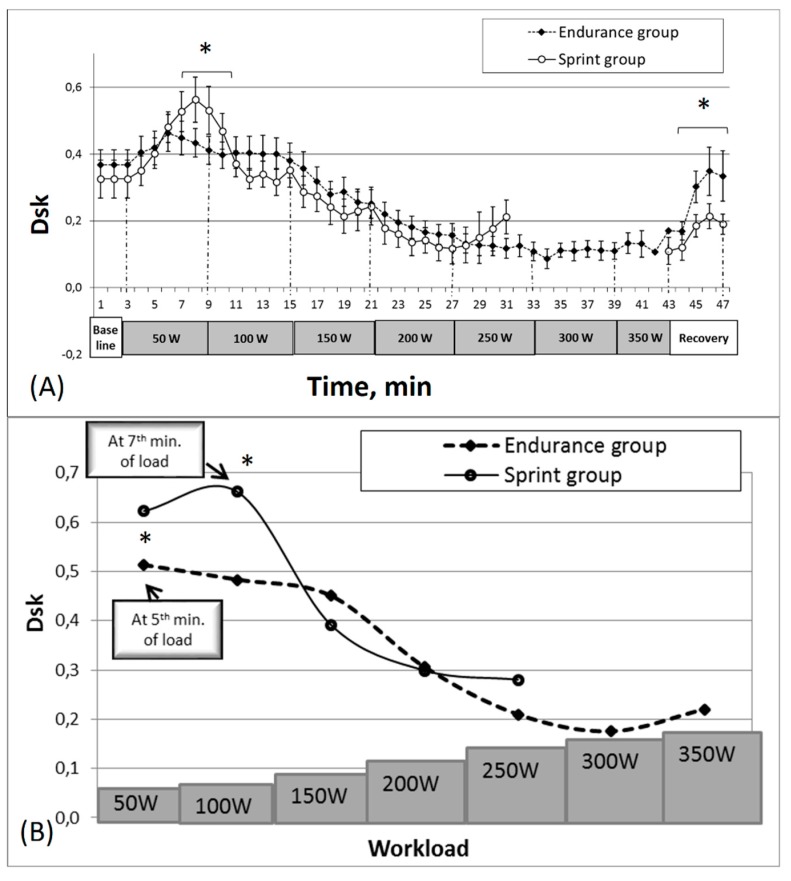
Average of values (**A**) and maximal values (**B**) changes of stroke volume and cardiac output dynamical interaction in sprint and endurance cohorts while performing incremental increase in workload every 6 min. Note: Axis x shows the study protocol, i.e., increase of workload in watts and from 44–47 min—the change of parameter during the recovery. * Statistically significant difference (*p* < 0.05).

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
