# Peer review of "Interaction between Cardiac Functional Indices during Incremental Exercise Test Reveals the Peculiarities of Adaptation to Exercising"

_medicina, 2019, doi:10.3390/medicina55070314_

Reviewer 1 Report

Major revisions:

 1)    Title: in my opinion it would be more appropriate to replace the term “Cardiovascular” with the term “Cardiac” since no vascular or cardiac-arterial coupling indices are analyzed.

2)    Material and Methods, page 2, line 76-78: I would provide these data in the Results section and in conjunction with significant/not significant differences between the two groups.

3)    Material and Methods, par 2.2: please specify how long the recovery time was.

4)    Material and Methods, par 2.3: please provide more details regarding the robustness of the adopted technique (reproducibility data available?). A picture of the experimental setup would be useful. Finally, here or in the discussion section I suggest to add consideration regarding the impact of stop exercising  on the final data (why did you take this decision? can this have an impact on the obtained results?).

5)    Material and Methods, par 2.: an explanation/consideration regarding the appropriateness of  applying algebraic co-integration method to these data would be useful.

Author Response

1. We take into account the suggestions of both reviewers and changed the title as recommended reviewer „Interaction between cardiac functional indices during bicycle exercise test reveals the peculiarities of adaptation to exercising“.

2. In manuscript we reported that there was no significant difference between the cohorts.

3. In manuscript we added that after the physical load the subjects rested and the cardiac parameter registered 5 minutes. Participants during all testing procedure were seated on a bicycle.

4. The study design was simple why we think that a few sentences are enough to describe this. We think that the additional figure of experimental protocol would be duplicated.

The participants decision to stop exercising was made by them if workloud was too hard to conitnue exercising. In only one case the decision to stop exercising was made by researchers because steep increase of  diastolic blood pressure was observed. According our concern this did not have impact on the resuts of this study.

5. In manuscript we explained that synergetic interactions between physiological mechanisms while exercising cannot described be by use statistical approach only, be-cause they based on averaged data analysis.

Reviewer 2 Report

I have problems to understand this paper: It is not only the english that causes it, but also the terminology.

It starts with the long title and the term interaction dynamics.

You want to show differences in the cardirespiratory system (heartrate , stroke volume ) of sprinters and endurance ahletes doing a bicycle test with increasing load.

In my opinion you can delete this term in your long title and in text all over.

You have to specify the workload as bicycle exercise test with increasing workload untill exhaustion.

Suggestion for title:

Differnces between sprint and endurance athletes in cardiocvascular function during a bicycle exercise test

Author Response

1. Manuscript has been submitted to ProofReadingService.com for editing and proofreading and editor has corrected the document, ensured consistency of the spelling, grammar and punctuation, and checked the format of the sub-headings, bibliographical references, tables, figures etc. We could add a certificate that the manuscript was corrected. If the reviewer concerns English language needs to be corrected anyway we can send and ask to edit the manuscript again.

2. We take into account the suggestions of both reviewers and changed the title as recommended reviewer „Interaction between cardiac functional indices during bicycle exercise test reveals the peculiarities of adaptation to exercising“.

Round  2

Reviewer 1 Report

Minor revision

1)    Material and Methods, par 2.3: please provide more details regarding the robustness of the adopted technique (reproducibility data available?)

2) Material and Methods, par 2.4: please provide literature reference for the added explanation regarding the necessity of algebraic cointegration method for these data.

Author Response

Thanks to reviewers for the valuable remarks. We have made the following improvements and changes in the manuscript:

1. We can not cleary undestand this remark.

In case the question is about method rheography, we presented the references about the measurements and calculations as to obtain the values of stroke volume and cardiac output.

In case the question is about the repeatability of the measurements, we should make reference on the data presented in Figure 2. The 6 repeated measures at rest and in each exercising stage of workload shows that measured values of stroke volume and cardiac output were almost the same until the fatigue has not occured. So these measurements are reproduceble.

2. We changed the structure of sentence to : „Statistical approach based on averaged data analysis cannot describe the synergetic interactions between physiological system and mechanisms while exercising“ and provided literature reference for the added explanation regarding the necessity of algebraic cointegration method for these data. 

Reviewer 2 Report

In the title bicycle exercise is included, but not in abstract and also too late in methods and in figures. Just writing about workload is not informative, because the chosen bicycle workload is not specific for sprinting and endurance runners. Include a word about this.

Author Response

Thanks to reviewers for the valuable remarks. We have made the following improvements and changes in the manuscript:

1. We agree with opinion that title must be changed. We change the word bicycle to word incremental.

2. In abstract we included that the bicycle ergometric method of incremental increase in a provocative workload (graded stress) was used.
